

# LocustLens: leveraging environmental data fusion and machine learning for desert locust swarm prediction

Sidra Khan[1], Beenish Ayesha Akram[1], Amna Zafar[2], Muhammad Wasim[3], Khaldoon S. Khurshid[2] and Ivan Miguel Pires[4]

[1] Department of Computer Engineering, University of Engineering and Technology Lahore, Lahore, Pakistan
[2] Department of Computer Science, University of Engineering and Technology Lahore, Lahore, Pakistan
[3] Department of Computer Science, University of Management and Technology, Lahore (Sialkot Campus), Pakistan
[4] Instituto de Telecomunicações, Escola Superior de Tecnologia e Gestão de Águeda, Universidade de Aveiro, Águeda, Portugal

Corresponding author
Ivan Miguel Pires, impires@ua.pt

## ABSTRACT

The desert locust is one of the most destructive locusts recorded in human history, and it has caused significant food shortages, monetary losses, and environmental calamities. Prediction of locust attacks is complicated as it depends on various environmental and geographical factors. This research aims to develop a machine-learning model for predicting desert locust attacks in 42 countries that considers three predictors: soil moisture, maximum temperature, and precipitation. We developed the Global Locust Attack Database for 42 countries (GLAD42) by integrating TerraClimate's environmental data with locust swarm attack data from the Food and Agriculture Organization (FAO). To improve the usability of spatial data, reverse geocoding which is the process of converting geographic coordinates (longitude and latitude) into human-readable location names (such as countries and regions) was employed. This step enhances the clarity and interpretability of the data by providing meaningful geographic context. This study's initial dataset focused on instances where locust attacks were recorded (positive class). To ensure a comprehensive analysis, we also incorporated negative class instances, representing periods (specific years and months) in the same countries and regions where locust attacks did not occur. This research utilizes the benefits of lazy learners by employing the K-nearest neighbor algorithm (K-NN), which provides high accuracy and the benefit of no time-consuming retraining even if real-time updated data is periodically added to the system. This research also focuses on building an eco-friendly machine learning model by evaluating carbon emissions from ML models. The results obtained from LocustLens are compared with other machine learning models, including baseline–K-NN, decision trees (DT), Logistic regression (LR), AdaBoost Classifier, BaggingClassifier, and support vector classifier (SVC). LocustLens outperformed all competitors with an accuracy of 98%, while baseline-K-NN achieved 96%, SVC gave 91%, DT gave 97%, AdaBoost has accuracy of 91%, BaggingClassifier gave 94% and LR gave 83%, respectively. Carbon emissions from RAM and CPU electricity consumption are measured in kg gCO2. They are a minimum for AdaBoost Classifier equal to 0.02 and 0.07 for DT and a maximum of 9.03 for SVC. The carbon footprint of LocustLens is 4.87 kg gCO$_2$.

## INTRODUCTION

The desert locust (Schistocerca gregaria) has been known to destroy crops across the globe throughout history. The desert locust pandemic began in 1986 and spread fast throughout the Sahelian nations in 1987, and reached northwest Africa by the end of that year. It spread to North Africa, the Sahel, Sudan, the Middle East, and Southwest Asia in 1988. Swarms then crossed the Atlantic to the Caribbean in October of that year (*Skaf, Popov & Roffey, 1990*). Desert locust plagues have ravaged Western Africa for centuries (*Gómez et al., 2019*), causing severe effects on the atmosphere (*Retkute et al., 2021*), food scarcity, crop losses, and affected the economies of several countries. Around twelve significant pest species of locusts and grasshoppers exist, allowing them to embark on extensive migrations and causing significant harm to crops, pastures, and other green vegetation during their swarming phase. According to estimates released by the United Nations' Food and Agriculture Organization (FAO) in March 2020, damages and losses in Yemen and East Africa alone total up to US$8.5 billion (*World Bank, 2020*). Nutrition, healthcare, and education are ignored when impacted households and families struggle to meet necessities like food. In Pakistan, a damage level of 15% to the output of wheat, gram, and potato alone was recorded, and losses to agriculture, in general, might reach a total of PKR 205 billion (USD 1.3 billion) in the future, reported by FAO (*Notezai & Rehman, 2020*). If swarm growth is unchecked, the World Food Program predicts that long-term reaction and recovery expenditures could exceed US$1 billion (*World Bank, 2020*). The devastation caused by locusts in 2020 and COVID has put much stress on already affected food output. Any future desert locust outbreaks can endanger 20% of the global output of the crops, risking famine regionally or on a global scale.

According to studies, a new generation of locusts emerges every eight weeks. Locusts exhibit a unique characteristic setting them apart from other insects: their population can rapidly expand, forming dense bands and swarms. During their solitary phase, locusts play a significant role in ecosystems. Nevertheless, shifts in environmental conditions and population growth can trigger the transition to their gregarious phase, potentially resulting in an outbreak. Every generation, the population of locusts increases by a factor of 20 on average (*World Bank, 2020*; *Symmons & Cressman, 2001*). The right combination of weather, soil, and vegetation is necessary for the outbreak as these factors encourage the reproduction and aggregation of formerly solitary individuals. The presence of rain makes it easier for locusts to reproduce. The soil environment is also one of the critical factors affecting locust reproduction and outbreaks (*Shuang et al., 2022*). Most of the countries and states affected by locust attacks are third-world countries. The Peninsula, Pakistan, Saudia Arabia, Africa, and India were affected extensively, resulting from the combination of warm weather conditions, uncharacteristically heavy rainfall, and inadequate monitoring

practices. Thus, COVID and locust attacks have caused massive destruction to the food and agriculture industry in history (*World Bank, 2020*).

Therefore, extensive research has been done on the prevention and early prediction of locust attacks. Effective management and control of the locust population is crucial due to the considerable damage they pose. The multidimensional nature of managing the locust population calls for a multidisciplinary approach. In the habitats of locusts, environmental changes (such as changes in land use) and weather unpredictability can produce ideal conditions for locust reproduction. It must be recognized and controlled at the appropriate time. Without these modifications, a locust outbreak could be triggered by an increase in population, causing the species to go from the solitarious to the gregarious phase (*Klein, Oppelt & Kuenzer, 2021*).

Researchers have extensively used machine learning models to predict locust breeding grounds, classify locust species, predict locust attacks, and analyze important factors that lead to invasion (*Tabar et al., 2021*). In this research, we have proposed a novel methodology called 'LocustLens', which accurately predicts the presence of desert locust swarms thereby mitigating risks to the agricultural sector by providing sufficient planning time. Such an early warning system can significantly impact various sectors, particularly agriculture, resulting in reduced losses and more effective management strategies. We have analyzed data from 42 countries considering the starting year of attack in different regions and environmental factors such as soil moisture, precipitation, and maximum temperature. Figure 1 is a pictorial representation of the top 20 out of 42 countries considered in this research. It is imperative to emphasize that the dataset accessible on the FAO website spans from 1985 to 2020, with the most recent update being documented in 2020. Because of this and the dataset's temporal scope, as well as the most current update accessible on the FAO platform, the data for 2021 to 2023 are not included in this research.

## CONTRIBUTIONS

### Data fusion

Extensive work has been done in predicting locust breeding grounds and their attacks (*Kimathi et al., 2020*). All these earlier methods share specific characteristics. For most of the research in this field, the FAO desert locust database (*Food and Agriculture Organization (FAO), 2023*) serves as the primary data source, emphasizing various regions of Africa. In this research, a comprehensive data collection process is carried out, collecting data for 42 countries related to locust swarm attacks and corresponding start year and environmental data from TerraClimate (*TerraClimate, 2023*). Data fusion combines positive values of locust attacks from FAO and key environmental factors from TerraClimate datasets. TerraClimate and FAO datasets are reverse geocoded to convert longitude and latitude to country names and regions. Moreover, negative classes are added for months where locust attacks were not reported to make the dataset favorable for predictions. The resulting dataset contains essential environmental factors strongly associated with locust attacks and geographical details of these 42 locust-prone countries, including the years and months when locust attacks typically begin. This data fusion resulted in a more intricate and comprehensive dataset named as GLAD 42 (Global Locust Attack Database for 42

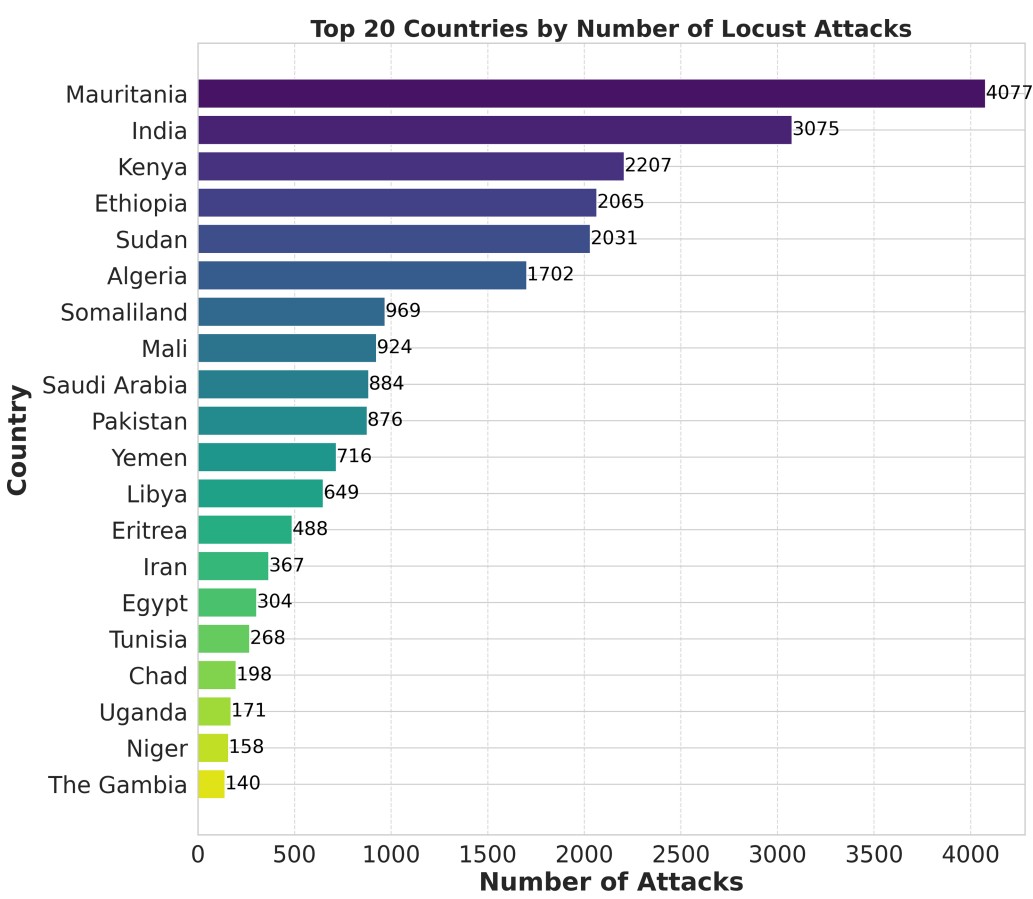

**Figure 1** Top 20 countries with locust attack count in GLAD42 (1985–2020).

countries) that includes all the vital independent features required for constructing a robust classification model.

## Use of lazy learners

Research in this field has three main areas: assessing locust risks, predicting breeding grounds, and forecasting locust presence. Multiple methods, such as Random Forest (RF), and neural networks along with other classification and regression algorithms, have successfully predicted African locust breeding grounds (*Gómez et al., 2018*; *Gomez et al., 2021*). As the literature review delves into, the classification research that has been done so far on locust attacks is either restricted to limited geographic regions, like Saudi Arabia, Kenya, and Africa, or it has only taken into account one or two environmental characteristics, primarily soil moisture, using remote sensing. However, a significant gap remains as broader global perspectives and comprehensive environmental factors have not been adequately addressed in the existing literature. Moreover, the majority of these studies have relied on FAO datasets, highlighting the need for more detailed and region-specific datasets encompassing a wider range of environmental factors. We have proposed LocustLens, using K-nearest neighbor (K-NN) following the renowned KISS

principle as the Lazy Learner methodology to enhance our ability to predict locust attacks across 42 countries. The KISS principle is a design principle that states that most systems function best when they are maintained simply as opposed to convoluted. Significant benefits from this approach, such as those seen with Lazy Learners like K-NN, are their simplicity, making them easy to understand and apply. Furthermore, their adaptability is a notable benefit. Lazy Learners quickly adapt to changes in the dataset due to their inherent nature, eliminating the need for time-consuming retraining and re-testing methods. Our algorithm adjusts fluidly to these updates as we add new country-specific data, guaranteeing effective scaling and enabling us to quickly add unique data inputs for accurate and reliable predictions of locust attacks.

## Novel methodology

We have proposed a novel methodology for building a classification model. In this novel approach, called LocustLens, we hierarchically employed the K-NN algorithm. It starts with stratified sampling, splitting the dataset into training and testing subsets. It then iterates over test samples based on the dataset creation; data subsets are created based on a country match of the test sample with corresponding fitting data while meticulously tracking the time involved. These subsets are smaller than the complete dataset, providing the performance advantage of the divide-and-conquer approach. K-NN is applied to these refined subsets for fitting. The fitted K-NN model predicts outcomes for test samples. Finally, the model computes average fitting and testing time and generates comprehensive performance evaluation results, providing insights into LocustLens's performance in predicting locust attacks across various countries.

## Sustainable machine learning

Existing studies have shown that desert locusts can change their behavior, ecology, and physiology in response to changes in climatic conditions (*Symmons & Cressman, 2001*). In our research paper, along with the prediction of locust attacks, we have examined the environmental impact associated with our model's fitting and testing phases. Machine learning models can affect our climate in various ways, some of which may be subtle. Potential emissions from machine learning (ML) models are divided into three categories: (1) system emissions, (2) application emissions, and (3) computing-related emissions. Specifically, we investigated the carbon footprint generated during computational processes, considering the total electricity consumed during training and testing and how much emissions result from CPU and RAM. This study is motivated by the significant role that greenhouse gas (GHG) emissions play in contributing to climate change, which subsequently influences factors such as precipitation patterns and temperature, which are crucial determinants in locust infestation. Figure 2 represents that embodied and operational emissions comprise the computing-related climate impact of machine learning models. The arrow indicates a direct relationship between the total amount of greenhouse gas (GHG) emissions and the embodied emissions connected to computing. Stated differently, the hardware utilized in machine learning models has an environmental impact and consumes energy, which leads to a rise in greenhouse gas emissions. On the other

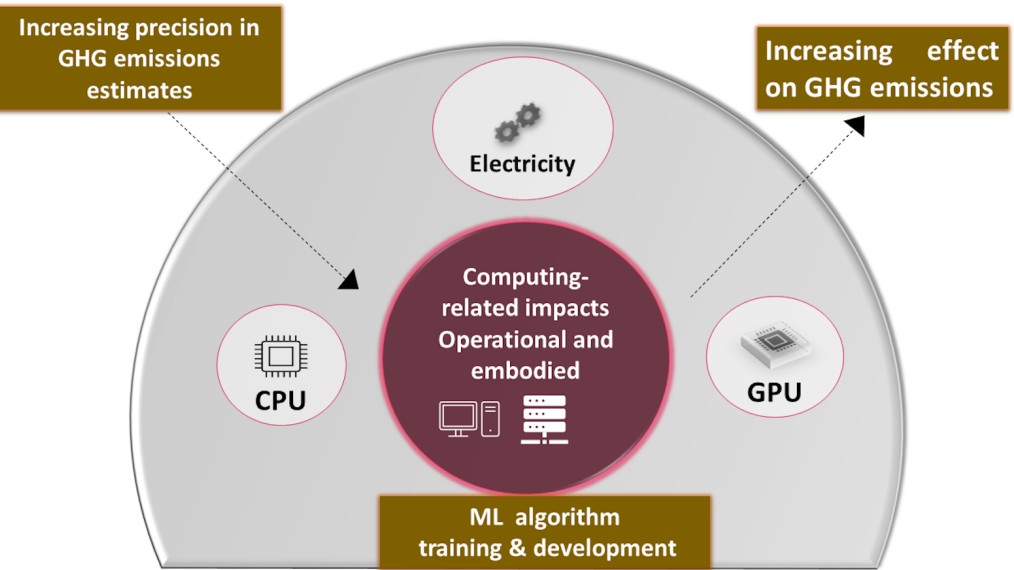

**Figure 2** Potential emissions from training and testing ML models.

hand, accuracy and precision in measuring or computing the environmental impact are what precision means in estimating GHG emissions. There is a greater understanding of the true environmental impact of the computing infrastructure when the estimations are more accurate.

## LITERATURE REVIEW

Swarms of locusts endanger the food sources of millions of people globally. As a result, research in this field was initiated a century ago by Boris Uvarov (*Pflüger & Bräunig, 2021*). Affected areas from locust attacks are the most fragile ones. Therefore, before digging into the details of building preventive systems, it is important to understand climate changes and ecological factors that trigger this attack. Other research has been done in many domains about desert locusts, from biological to psychological, financial, and many more aspects. However, we are more concerned with the management and control aspect of the desert locust. Research in this field can be broadly categorized into three sub-fields:

1. Research related to the desert locust infestation and the reasons behind the outbreak.
2. Predicting breeding grounds of the desert locust
3. Predicting the presence of desert locust and its swarms.

During the recent desert locust outbreak in 2020, when COVID-19 also caused massive destruction, researchers aimed to identify the risks posed by the outbreak to people from different socioeconomic classes and their fields of work. Researchers analyzed the surface soil moisture data, soil type, temperature data, and wind trajectory to predict the high-risk areas of a desert locust invasion in India (*Ghosh & Roy, 2020*). In *Latchininsky (1998)*, they have discussed climate changes, such as annual precipitation changes in deserts, which directly affect the population of locusts. Using preventive measures, spraying pesticides was

considered the principal method to control locust attacks (*Showler et al., 2021*). However, it has severely threatened human health and the environment (*Nicolopoulou-Stamati et al., 2016*).

Moreover, some researchers proposed using modern technology, for example, unnamed aerial vehicles (UAV) or drones, to survey remote areas for effective outbreak control (*Matthews, 2021*). As the world recovered from the outbreak, researchers analyzed the factors that led to the outbreak in 2020 using soil moisture, normalized difference vegetation index, and digital elevation model. They discussed anomalies of these variables that led to the outbreak (*Wang et al., 2021*).

Machine learning is used for its accurate results and reliability in this domain. Researchers have tried to make early warning systems using different methodologies to predict the presence of desert locusts or their swarms. RF has been used a lot, considering soil moisture from remote sensing as a critical factor to predict the presence of locusts, providing an area under the curve (AUC) of 0.761 (*Cyril Piou et al., 2019*). A data-driven forecast model using support vector machines (SVM) has been designed with indicators like the land cover, topography, vegetation, soil, and similar factors for predicting locusts' presence and breeding, and it gave an accuracy of 77.4% and an F-score close to 0.772 (*Sun et al., 2022*). *Kimathi et al. (2020)* used NOAA, ISRIC, and WorldClim rainfall and temperature data with a Maximum Entropy (MaxEnt) model to identify locust breeding sites in East African countries. NOAA's CPC Soil Moisture Data provided crucial insights into soil conditions, while ISRIC World Soil Information offered global soil property data relevant to locust breeding. WorldClim contributed high-resolution climate data, including rainfall and temperature, which are critical for locust habitat modeling. These datasets were essential in understanding the environmental factors influencing locust outbreaks in the region. MaxEnt model uses the concept of maximum entropy to find the probability of the existence of locusts in distributed space. They next used the trained model to apply to the other countries in their dataset to verify the accuracy of their model. They achieved AUC of 0.887, 0.884, and 0.820 for Mauritania, Kenya, and Saudi Arabia, respectively.

Moreover, locust classification and crop production analysis using ResNet50 have been done by *Ye et al. (2020)*, where they used the concept of Densenet to help farmers identify desert locust locations and predict their attack. To increase the network's stability, convergence speed, and classification precision, they integrated the BatchNorm function before each convolution layer and obtained an accuracy of about 90.16%. *Samil et al. (2020)* effectively applied a basic long short-term memory (LSTM) model to forecast swarm locations one month ahead of time based on historical data from all impacted regions, achieving recall and precision of 81% and 60%, respectively. A deep learning method utilizing LSTM and convolutional neural networks (CNN) used in *Tabar et al. (2021)* for forecasting future challenges of desert locusts in Africa. It suggests PLAN, a machine learning system for forecasting high-resolution spatial and temporal locust movement. PLAN uses a special crowd-sourcing dataset, remote-sensed environmental data, and a modular neural network architecture to produce precise forecasts of locust migration. Two species that are common in Zambia are the red locust (Nomadacris

septemfasciata) and the African migratory locust (*Locusta migratoria migratoriodes*), which were detected using CNN with the precision of 91% and 85% (*Halubanza et al., 2022*).

Incorporating imprecise and unclear information into the prediction of locust attacks is made possible by fuzzy logic, which allows for a more flexible and realistic modeling of the elements driving locust behavior. It improves the predictability of outcomes in dynamic and intricate ecological scenarios by enabling the system to manage variables with degrees of truth. Therefore, researchers used fuzzy logic to study the timing of the hatching of locusts as part of their efforts to respond early to desert locust swarming in eastern Africa (*Landmann et al., 2023*) and achieved an accuracy of 82%.

Remote sensing plays a crucial role in locust preventive management, aiding in the mapping and monitoring of extensive locust habitats. This proactive approach aims to identify and control locust population growth at a smaller scale before they escalate into larger-scale plagues (*Klein, Oppelt & Kuenzer, 2021*; *Hunter, 2004*). *Klein et al. (2022)* used RF for time series analysis of Moroccan locusts in Sardinia, focusing on remote sensing data, which proved helpful in risk assessments and predictions of locusts. For binary cropland classification, they achieved an accuracy of 96.4% and a kappa coefficient of 0.951 (*Klein et al., 2022*). Deep learning and computer vision algorithms have proved to be highly effective in detecting locust presence and analyzing damage caused by their infestation. In *Karim (2020)*, they used Resnet50 and MobileNet to train on the ImageNet dataset for rice disease detection and pest recognition. They have achieved an accuracy of 73.34% and 76.45%, respectively. Along with the detection of crop losses using image segmentation, the detection of locusts using image processing and their classification has been done by *Ebrahimi et al. (2017)* and reported an error rate of less than 2.5%. Ensemble techniques have consistently shown improvement in performance across a combination of datasets. *Santana et al. (2014)* have developed a classification model for bees using MLP, reporting an accuracy of 87.68%. In *Kasinathan, Singaraju & Uyyala (2021)*, researchers have used artificial neural network (ANN), CNN, K-NN, support vector machine (SVM), and naïve Bayes for nine-class and 24-class datasets. The highest classification rate of 91.5% and 90% was reported for CNN. Along with SVM and RF, lazy learners have proved more straightforward and more robust in classification. Therefore, in the latest study in 2023, researchers have used K-NN as a classifier in pest prediction and achieved an accuracy of 99.3% (*Pusadan & Abdullah, 2022*). Moreover, several crop pests have been predicted using time-series feature extraction and transfer learning. This combination resulted in enhanced accuracy of prediction for crop pests. They have used Pearson product-moment coefficient for machine learning models including K-NN, RF, SVM, and naïve Bayes and achieved the highest accuracy of 0.9661 with RF in the prediction of aphid pests and 0.944% in the prediction of whiteflies (*Tsai et al., 2023*). *Cornejo-Bueno et al. (2023)* in Western Africa related to desert locusts has demonstrated regression and classification techniques in the context of locust sightings. Various machine learning algorithms, including DT, RF, SVM, and multilayer perceptron (MLP), were applied to enhance the accuracy of desert locusts' presence prediction, resulting in improved precision and recall (*Cornejo-Bueno et al., 2023*). The related work we have considered in this study is summarized in Table 1.

**Table 1** Literature review summary.

| Paper | Dataset | Model | Outcome | Metrics |
|---|---|---|---|---|
| *Piou et al. (2019)* | Proprietary | RF | Locust presence Forecasting | AUC: 0.761 |
| *Sun et al. (2022)* | FAO (*Food and Agriculture Organization (FAO), 2023*) | SVM | Locust presence Forecasting | AUC: 0.766, Accuracy: 77.46% |
| *Kimathi et al. (2020)* | i. Worldclim2 (*WorldClim (2023)* ii. NOAA (*NOAA Climate Prediction Center (CPC), 2023*) iii. ISRIC *Information (2023)* | MaxEnt | Locust breeding ground prediction | AUC: 0.887 |
| *Ye et al. (2020)* | Proprietary | Resnet-locust-BN | Locust species Classification | Accuracy: 93.50% |
| *Samil et al. (2020)* | FAO (*Food and Agriculture Organization (FAO) (2023)* | LSTM | Prediction of swarm's location | Precision: 60% Recall: 81% |
| *Tabar et al. (2021)* | Proprietary | CNN+LSTM +FFN | Forecasting migration patterns | Accuracy: 0.830% |
| *Halubanza et al. (2022)* | Proprietary | CNN+MobileNet | Locust species Classification | Precision: 91%, 85% |
| *Landmann et al. (2023)* | Proprietary | Fuzzy logic | Prediction of locust Hatching | Accuracy: 82% |
| *Klein et al. (2022)* | Proprietary | Random Forest | Locust Prediction | Accuracy: 96.4% |
| *Karim (2020)* | Proprietary | ResNet50 + MobileNet | Pest species Classification | ResNet Accuracy: 73.34% MobileNet Accuracy: 76.64% |
| *Ebrahimi et al. (2017)* | Proprietary | SVM | Pest species Classification | MAE = 2.25% |
| *Santana et al. (2014)* | Proprietary | MLP | Pest species Classification | Accuracy: 87.68% |
| *Kasinathan, Singaraju & Uyyala (2021)* | Proprietary | CNN | Insect species Classification | Accuracy (nine-class dataset): 91.5% Accuracy (24-class dataset): 90.5% |
| *Pusadan & Abdullah (2022)* | Proprietary | K-NN | Pest prediction | Accuracy: 99.3% |
| *Tsai et al. (2023)* | i. Taiwan's agriculture (*Central Weather Bureau of Taiwan (CWB), 2023*), ii. Crop pest DSS (*Indian Council of Agricultural Research, ICAR)(2023)* | KNN+RF+SVM + Naïve bayes | Insect species Classification (white fly) | Accuracy: (KNN): 0.95%, (RF): 0.94%, (DT): 0.86, (SVM): 0.84% |
| *Cornejo-Bueno et al. (2023)* | FAO (*Food and Agriculture Organization (FAO), 2023*) | SVM+DT+RF | Locust species Classification | AUC: (RF): 0.92, (DT): 0.86, (SVM): 0.88 |

**Notes.**
    *Locust Forecasting implies forecasting of locust occurrence, their breeding regions, and migration patterns in areas under study.
    **Locust classification: Classification of different locust's species and prediction of their attack.
    ***Locust Prediction implies a prediction of locust attacks in regions under study.

# DATASET GLAD _42 DESCRIPTION

This research presents a novel dataset aimed at enhancing the accuracy and precision of machine learning models for predicting locust breeding and attacks. The dataset integrates multiple sources, focusing on environmental factors such as maximum temperature, precipitation, and soil moisture, which are crucial in predicting locust activity, alongside locust occurrence data from 42 countries spanning from 1985 to 2020.

## Data sources and integration process
### *Desert locust swarms global watch data:*
The primary source of locust occurrence data is the FAO's Desert Locust Swarms Global Watch dataset, which provides detailed time series data from 1985 to 2020 (*Food and Agriculture Organization (FAO), 2023*). This dataset includes critical features such as the start year, start month, and geographical coordinates (latitude and longitude) of locust attacks. The FAO dataset serves as the backbone for locust occurrence records, offering precise temporal and spatial data on locust activities globally.

### *TerraClimate dataset*
The environmental data was derived from the TerraClimate dataset (*TerraClimate, 2023*), which combines high-spatial-resolution data from the WorldClim dataset with coarser-resolution datasets like CRU Ts4.0 and JRA-55. WorldClim provides high-resolution, gridded climate data, including long-term averages of temperature, precipitation, and other key environmental variables, widely used for ecological and environmental research due to its detailed spatial coverage. In this study, WorldClim data is crucial for assessing the impact of climate factors like temperature and precipitation on locust behavior and outbreaks. TerraClimate data is extracted from WorldClim and it offers a comprehensive monthly dataset that includes variables such as maximum temperature, vapor pressure, accumulated precipitation, runoff, and soil moisture, spanning from 1958 to 2020 with a spatial resolution of 1/24° (4 km) (*Abatzoglou et al., 2018*; *TerraClimate, 2023*). For this study, the focus was on three key environmental features—soil moisture, maximum temperature, and precipitation—during the period 1985 to 2020, aligning with the FAO locust occurrence data. The Fig. 3 shows relationship between environmental features and target variable and correlation among them.

### *Data processing and feature engineering*
The raw data from TerraClimate was initially in netCDF format, which required conversion to a more accessible format for integration with the FAO dataset. Three features including soil moisture, maximum temperature, and precipitation conversion of TerraClimate files which span from 1985 to 2020, are done as they were available in netCDF format. The longitude and latitude of countries are reverse geocoded to convert into region and country names. Figure 4 shows the complete data collection and fusion process.

To ensure temporal alignment, the FAO dataset's locust occurrence records were integrated with the corresponding environmental data from TerraClimate for the same years and months. This integration was carefully managed to maintain the integrity and accuracy of the temporal and spatial dimensions of the data. Hence, our final dataset has seven independent features and one dependent feature with class labels according to locust presences as "yes" and "no". This dataset is used in building a binary classifier for locust attack prediction.

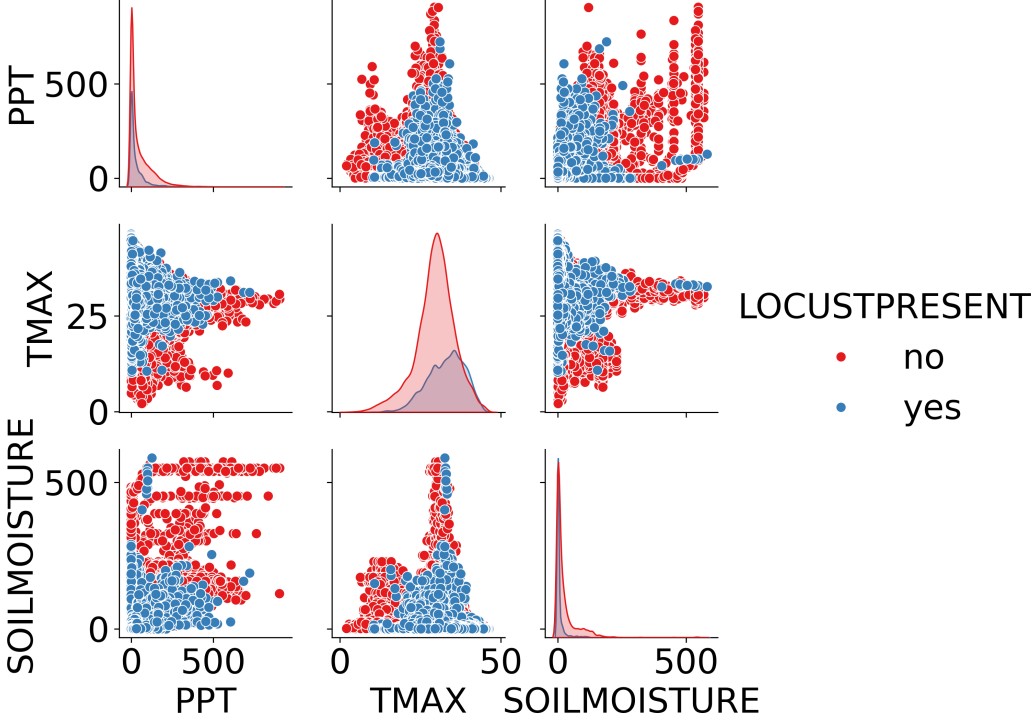

**Figure 3** Pair plot showing the relationship between environmental factors (**precipitation, maximum temperature, and soil moisture**) **and Locust presence.** The plot highlights the distribution and correlation of these variables, differentiating between areas with locusts (blue) and without locusts (red).

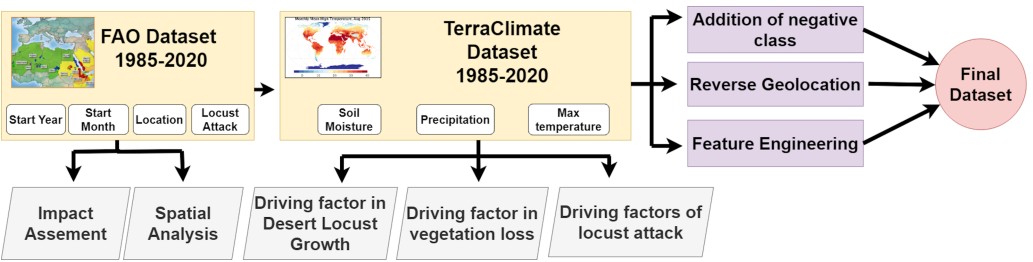

**Figure 4** **GLAD42 generation process.**

### *Dataset statistics and summary*

GLAD42 is designed to support the development of a binary classifier for locust attack prediction. The environmental data, such as precipitation, maximum temperature, and soil moisture, cover a wide range of values.

The dataset has precipitation values in mm ranging from 0.0 to 906.7, the maximum temperature in Celsius ranging from 2.3 to 46.71, and soil moisture in VWC in the range of 0 to 569.5 for 42 countries along with specific latitude, longitude, start year, and start month. The statistics for three environmental features considered in this study are shown in Table 2. This extensive range reflects the diverse ecological conditions across

**Table 2  Summary statistics for environment features.**

| Statistics | Precipitation | Maximum temperature | Soil moisture |
|---|---|---|---|
| Count | 79,528 | 79,528 | 79,528 |
| Mean | 46.48 | 30.56 | 23.05 |
| Std Dev | 71.67 | 6.21 | 50.55 |
| Min | 0.00 | 2.10 | 0.00 |
| 25% | 1.00 | 27.10 | 0.10 |
| 50% | 15.70 | 30.76 | 2.90 |
| 75% | 64.80 | 34.72 | 21.70 |
| Max | 906.70 | 46.71 | 582.60 |

**Table 3  Dataset GLAD42 features.**

| Feature name | Description | Type |
|---|---|---|
| Start month | Month when desert locust swarms were reported | Independent |
| Start Year | Year when desert locust swarms were reported | Independent |
| Country name | Country name where swarms were reported | Independent |
| Region | Region name where swarms were reported | Independent |
| Soil moisture | Soil moisture of the area where swarms were reported | Independent |
| Precipitation | Precipitation of the area where swarms were reported | Independent |
| Max temperature | Maximum temperature of the area | Independent |
| Locust present | Target variable | Dependent |

the 42 countries studied, providing a robust foundation for predictive modeling. Table 3 summarizes GLAD42 features.

## METHODOLOGY

### Feature engineering

The latitude and longitude are converted to human-readable addresses using Reverse Geocoding. "Country name" and "Region" help in predicting the presence of desert locust swarms over a larger area and improve the readability of our dataset. We added a target variable called "Locust Present" (set to yes by default as the FAO dataset only contains positive class). We then selected the following Features: "Start Year", "Start Month", "Region", "Country name" (extracted from latitude and longitude *via* Reverse Geocoding), "Precipitation", "Soil Moisture", "Maximum Temperature", and "Locust Present". Data cleaning is done by dropping any row containing a null value. The data obtained from FAO exclusively contained records of countries where locust occurrences were reported, forming the foundation of our dataset. To ensure a more comprehensive and balanced dataset, we generated a negative class representing instances when desert locust swarms were absent. To provide a concrete example, consider India in the 1990s: In 1990, desert locust swarms were observed in October and November in just one out of the initially identified six regions. To address this absence, we included additional rows for that specific month. These additional rows included the Location Name, Start Year (1990), and

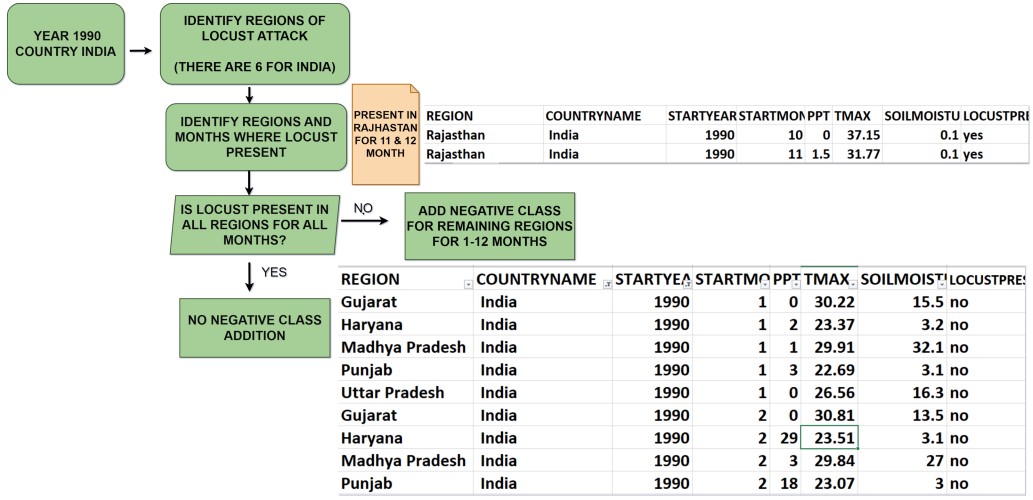

**Figure 5 Negative class generation.**

Start Month, indicating "no" for locust Presence in the remaining regions. We repeated this process every month, spanning from 1985 to 2020. Figure 5 is a visual representation of this process.

Following this data augmentation, we conducted rigorous data quality assessments and addressed missing values. The resulting dataset, comprising 79,528 rows and seven columns, is the foundation for our work. It allowed us to explore the temporal and spatial dynamics of desert locust swarm occurrences across various locations during the studied period with precision and accuracy. Figure 6 depicts country wise positive and negative instance. It is evident from the count that the number of instances for each country varies in terms of positive and negative class labels.

## Locustlens

LocustLens aims to predict locust attacks with high accuracy and time efficiency. In LocustLens, we have hierarchically used the K-NN algorithm. K-NN is a versatile and adaptable method for classification and regression tasks that can manage several data types.

LocustLens begins with dividing the dataset into training (N) and testing (M) sets using stratified sampling. For each test sample (M[i]), we created smaller subsets from the training data, focusing on samples from the same country as the test sample while keeping track of the time it takes. The K-NN model is fine-tuned using hyperparameters like the number of neighbors, weights, and distance metrics.

- In dataset feature space, point $x$ represents an individual test sample, denoted as M[i], for which we are making locust swarm prediction. This test sample belongs to a specific country.
- $y$ corresponds to the class labels of the training samples, either 'yes' or 'no' within the country-specific subset that matches the country of the test sample $x$.

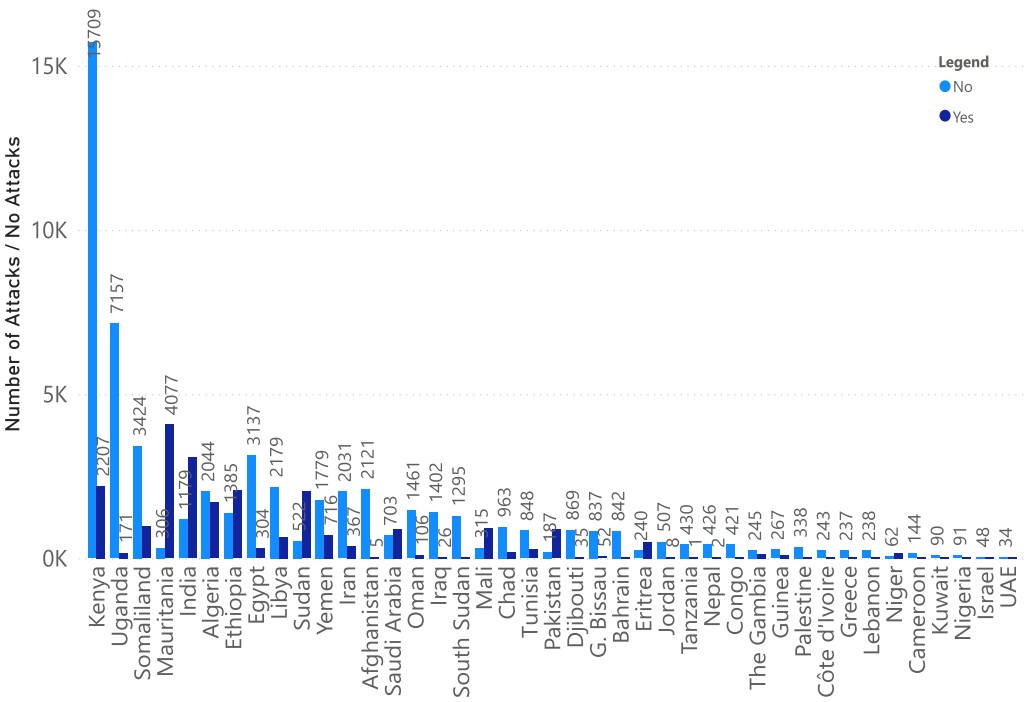

**Figure 6** **Positive and negative sample distribution country-wise.**

After fitting and testing LocustLens for different sets of hyperparameters the best optimal configurations were noted using randomized search. Equation (1) is used to calculate the distance to pick 'k' nearest neighbors.

$$\text{Manhattan distance} = \sum_{r=1}^{d} w_{r \times} |x_r - z_r| \tag{1}$$

where:

- $x$ represents test sample M[i]
- $d$ represents number of dimensions or feature space denoted by $r$; $w$ represents weights which in our case ($w_r = 1$ for all r).
- $z$ represents a training sample to which we are measuring distance.

Equation (2) is used to make predictions on data point $x$, and the model assigns class label C that occurs most frequently among the 'k' nearest neighbors to $x$ accordingly.

$$C(x) = \underset{C}{\text{argmax}} \sum_{i=1}^{k} \mathbb{I}(y_i = C). \tag{2}$$

- $\mathbb{I}(y_i = C)$ is the indicator function that is 1 if $y_i = C$ (*i.e.*, if the *i*th neighbor belongs to class $C$) and 0 otherwise. $C(x)$ is the class assigned to data point $x$.
- $y_i$ is the class label of the *i*th neighbor
- $\mathbb{I}(y_i = C)$ is the indicator function that is 1 if $y_i = C$ (*i.e.*, if the *i*th neighbor belongs to class $C$) and 0 otherwise.

**Table 4   LocustLens hyperparameters.**

| Parameter | Default | Tuned parameters | Description |
|---|---|---|---|
| n_neighbors | 5 | 7 | neighbors used to make decision. |
| weights (uniform, distance) | uniform | uniform | Weight function used in prediction |
| Algorithm (auto, ball_tree, kd_tree, brute) | auto | auto | 'auto' will attempt to decide the most appropriate algorithm. |
| Metrics (Minkowski, Euclidean, Manhattan) | minkowski | manhattan | Metric to use for distance computation. |

Hyperparameters selected after applying a randomized search algorithm on a set of hyperparameters for LocustLens are listed in Table 4.

Therefore, LocustLens is evaluated recursively for all countries in the dataset, resulting in a high accuracy score. LocustLens is developed using a lazy learner algorithm. We opted for this approach because lazy learners like K-NN are robust and well-suited for managing our extensive dataset covering 42 countries. This choice ensures that our model remains adaptable and responsive to evolving data conditions, maintaining accuracy and relevance as new data of countries and data of new countries are incorporated. Additionally, our model assesses the time-sensitivity of both the training and testing phases and measures carbon emissions resulting from the machine learning model. Thus, LocustLens is a highly flexible, accurate, and optimized predictive model for locust prediction. Figure 7 is a graphical representation of the working of LocustLens.

## Comparison models

A thorough review of the literature served as the foundation for comparing LocustLens with other models. Models like support vector classifier (SVC), logistic regression (LR), K-NN, and decision trees (DT) were found to be frequently used in the classification of pest or locust attacks on crops. These models were selected because they exemplify industry standards for binary classification problems including non-imagery data, which is comparable to the GLAD42 dataset utilized in this research. Making the comparison with these models is essential to comprehending LocustLens's performance in a proven setting. The investigation can show whether LocustLens provides any notable benefits or overcomes particular difficulties more effectively than by comparing it to these widely utilized techniques. Table 5 lists down all combinations of hyperparameters used in comparison with LocustLens.

## EVALUATION METRICS

### Classification metrics

The performance metrics used to evaluate models are explained in this section. These metrics are adaptable to scalar analysis, which provides a quantitative measure of model performance, and graphical analysis, which provides visual insights into model performance (*Tharwat, 2021*). The area under the receiver operating characteristic curve (AUC), F1-score, precision, recall, and accuracy are the scalar metrics we have considered for evaluation. A summary of these measures is provided below:

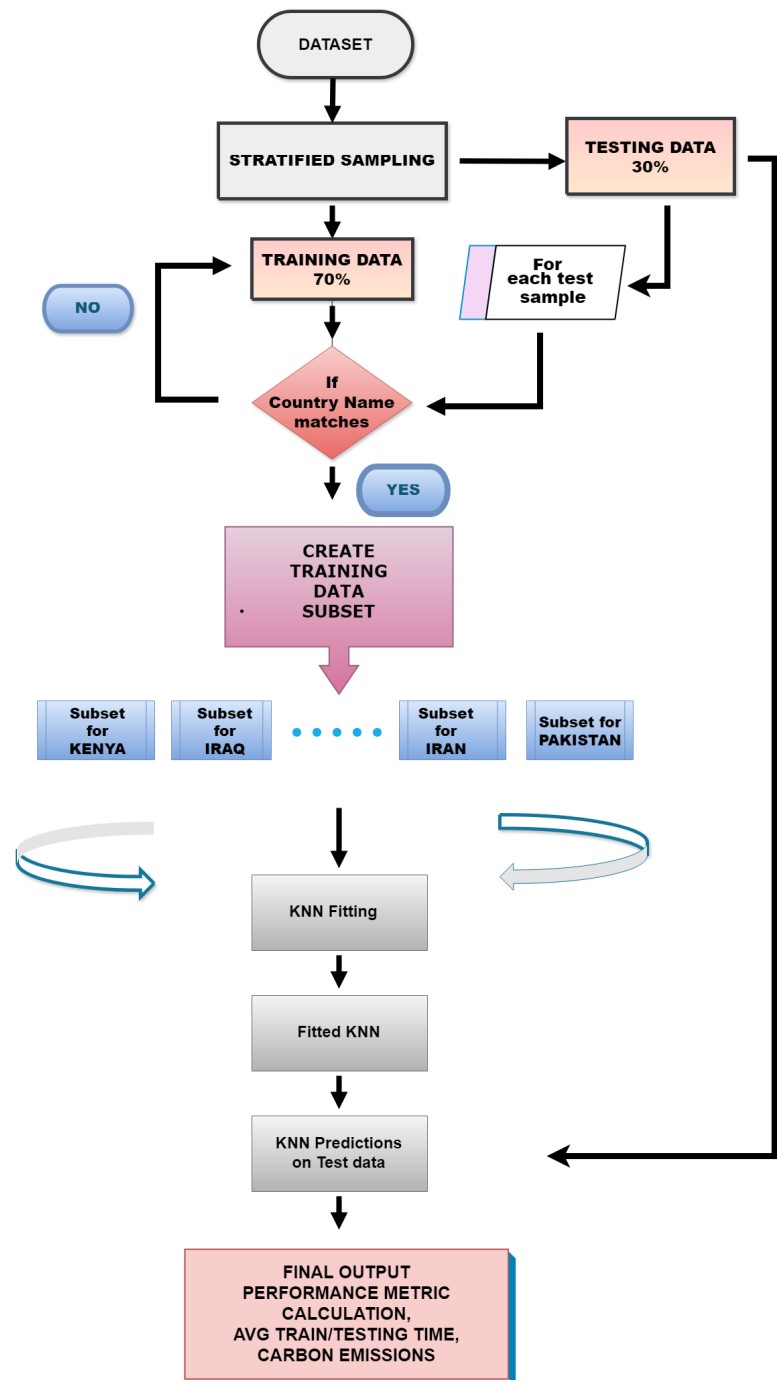

**Figure 7** Proposed methodology for LocustLens.

Values predicted by the classifier can be classified into four different labels and confusion matrix can be used to visualize them. Table 6 is depicting four classification labels that can be further used to measure accuracy, precision, recall, AUC and F1-score.

1. True positive (TP): when LocustLens predicted value is 'yes', and ground truth is 'yes'.

---

**Algorithm 1** LocustLens Algorithm

---

**Input:** $\{(x_i, y_i)\}$ **where** $i = 1$ **to** $N$
$x_i = \{\text{Region}_i, \text{CountryName}_i, \text{Start Year}_i, \text{Start Month}_i, \text{PPT}_i, \text{TMAX}_i, \text{soil moisture}_i\}$ $y_i = \{\text{Locust\_Present}_i\}$
**Output:**
$N$ data subsets
Trained K-NN
LocustLens Classifier

---

**Phase I:**
**Input = 7 independent features / 1 dependent feature, Output = Final Dataset**
    1. Independent features = Start Year, Start Month, Precipitation, Maximum temperature, soil moisture, X coordinates, Y coordinates,
    2. Dependent features = LocustPresent (class label = "yes")
    3. Reverse geocoding to get Region and Country name.
    4. Identify missing values.
    5. Feature Scaling of independent and dependent features
    6. Negative class addition to LocustPresent
The output of Phase I is the Final Dataset denoted as $\{(x_i, y_i)\}$ where $i$ ranges from 1 to $N$.

---

**Phase II:**
**Input = Final Dataset with input** $\{(x_i, y_i)\}$
    1. Initialize stratified sampling: splits, shuffle, and random state.
    2. for i: 1 to X and Y index
    3.    Get stratified $X_{\text{train}}$, $X_{\text{test}}$
    4.    Get stratified $Y_{\text{train}}$, $Y_{\text{test}}$
    5.    for j: 1 to $X_{\text{test}}$
    6.    Make a list: CountryName of j samples
    7.    Get $X_{\text{train}}$ indices where CountryName of jth sample $= X_{\text{train}}$ samples
    8.    Generate data subsets S based on CountryName
    9.    Initialize KNN: weights, neighbors, distance.
   10.    Fit KNN on S
   11.    Get prediction on each $X_{\text{test}}$ [j]
   12.    Record predicted vs Y values.
   13.    end for
   14. end for
   15. Record classification report (Accuracy, Precision, Recall, F1-Score, AUC)
   16. Record average training and testing time
   17. Record carbon emissions of training and testing of the model.

---

   2. True negative (TN): when the LocustLens predicted value is 'no', the ground truth is also 'no'.

   3. False positive (FP): when LocustLens predicted value is 'yes', and the ground truth is 'no'.

   4. False negative (FN): when the LocustLens predicted value is 'no', and the ground truth is 'yes.'

Based on the metrics mentioned above, accuracy, precision, recall, and F1-score can be calculated according to the Eqs. (3) to (6).

*Accuracy*: The ratio of correct predictions to total predictions.

$$\text{Accuracy} = \frac{TP + TN}{TP + TN + FP + FN} \tag{3}$$

*Precision*: It is the ratio of correctly predicted positive values to all the predicted positive values.

$$\text{Precision} = \frac{TP}{TP + FP} \tag{4}$$

**Table 5  Combined hyperparameters for various models.**

| Model | Parameter | Default | Tuned parameters | Description |
|---|---|---|---|---|
| | C | 1.0 | [0.1, 1, 10] | Regularization parameter. |
| SVC | kernel | rbf | ['linear', 'rbf'] | The kernel used in the algorithm. |
| | gamma | scale | [scale, auto] | Kernel coefficient for 'rbf', 'poly', and 'sigmoid'. |
| | Criterion | gini | ['gini', 'entropy'] | Function to measure split quality. |
| | max_depth | None | [None, 10, 20, 30, 1000] | Maximum depth of the tree. |
| Decision Tree | min_samples_split | 2 | [50, 100] | Minimum samples required to split an internal node. |
| | min_samples_leaf | 1 | [1, 2, 4, 10] | Minimum samples required at a leaf node. |
| | max_features | None | ['auto', 'sqrt', 'log2'] | Number of features considered for the best split. |
| | penalty | l2 | ['l1', 'l2'] | Norm of the penalty. |
| LR | C | 1.0 | [0.01, 0.1, 1, 10, 100] | Inverse of regularization strength. |
| | solver | lbfgs | ['liblinear', 'saga'] | Algorithm used in the optimization problem. |
| | max_iter | 100 | [100, 200, 300, 500, 1000] | Maximum iterations for solvers to converge. |
| | base_estimator | None | DecisionTreeClassifier | Base classifier for ensemble. |
| | n_estimators | 50 | 10 | Number of estimators for boosting. |
| AdaBoost | learning_rate | 1 | 1 | Contribution of each classifier. |
| | algorithm | SAMME.R | SAMME.R | Algorithm to find class probabilities. |
| | random_state | None | 42 | Seed value. |
| | base_estimator | None | SVC | Base classifier for ensemble. |
| | n_estimators | 10 | 10 | Number of estimators in the ensemble. |
| Bagging Classifier | learning_rate | 1 | 1 | Contribution of each classifier. |
| | verbose | 0 | 0 | Verbosity control. |
| | random_state | None | 42 | Seed value. |
| | n_neighbors | 3, 5,7 | 11 | Number of neighbors to use. |
| Baseline-K-NN | weights | uniform, distance | ['uniform', 'distance'] | Weight function used in prediction. |
| | p | 1,2 | 1 | Power parameter for the Minkowski metric. |

**Table 6  Confusion matrix.**

| | |
|---|---|
| True Positive (TP) | False Negative (FN) |
| False Positive (FP) | True Negative (TN) |

***Recall***: Recall/sensitivity is the ratio of correctly predicted positive values to all the actual positive values.

$$\text{Recall} = \frac{TP}{TP + FN} \tag{5}$$

***F1 –score***: It is the harmonic mean of precision and recall.

$$\text{F1-score} = \frac{2 \times \text{Precision} \times \text{Recall}}{\text{Precision} + \text{Recall}}. \tag{6}$$

***AUC***: When a model makes a prediction, it frequently gives each instance a probability score, indicating how likely it is to fall into the positive class. The model's capacity to differentiate between these two groups using these probability scores is measured by the AUC metric. The value can be between 0 and 1, with higher numbers indicating greater

model performance. An AUC of 1 denotes a perfect model, while an AUC of 0.5 denotes a model that performs no better than random guessing.

## Estimating carbon footprints

Locust attack results when the climate becomes favorable for their breeding. Machine learning models can delicately and variably affect the climate, affecting environmental dynamics in complex and direct ways. The environmental impact of ML models can be categorized into embodied emissions, which result from hardware production, and operational emissions, arising from ML model development, data processing, training, and inference. To quantify the carbon footprint of ML training, we employed the package called CodeCarbon (_CodeCarbon, 2023_), which calculates emissions based on the following factors:

$$\text{Carbon Emissions}(CO_2 - \text{equivalent}) = \text{Energy Consumption (kWh)}$$
$$\times \text{Regional Carbon Intensity (in } CO_2 \text{ per kWh).} \tag{7}$$

CodeCarbon tracks and records the power usage of GPU, CPU, and RAM during code execution by capturing data at regular 15-second intervals to measure electricity consumption accurately. The regional carbon intensity of electricity refers to the carbon emissions associated with generating electricity in a specific geographical area or region (_Google Cloud, 2023_). It is typically measured in terms of carbon dioxide $CO_2$ emissions per unit of electricity generated. Thus, using the same procedure, we will measure carbon emissions for our machine-learning models.

## RESULTS AND DISCUSSION

The findings of this research are discussed in this section, focusing on the performance and comparison of LocustLens, a model constructed using the K-NN algorithm with a divide-and-conquer approach. This method partitions the dataset for better performance in predicting locust attacks, and its accuracy is evaluated against several baseline models, including DT, SVC, Bagging, Boosting, and LR. From the literature (_Pusadan & Abdullah, 2022_; _Piou et al., 2019_; _Sun et al., 2022_; _Santana et al., 2014_), it is evident that machine learning models such as K-NN, SVC, RF, DT, and LR are frequently used for insect and locust attack prediction tasks. However, our methodology using LocustLens differs from existing approaches, particularly _Pusadan & Abdullah_'s (_2022_) work which achieved 99.3% accuracy. The key differences lie in data type and distance metrics: we use non-image data and the Manhattan distance for robustness, whereas Pusadan used image data and Euclidean distance. Additionally, instead of applying K-NN to the entire dataset, LocustLens applies the divide-and-conquer strategy using country-specific data, improving prediction precision across countries. In our experiments, we initially implemented a simple K-NN model while fitting and testing on dataset with various hyperparameters. To improve baseline-K-NN we developed LocustLens which employed country-wise fitting and testing instead of fitting on the entire dataset. Hyperparameter optimization was achieved through randomized search, in contrast to Pusadan's cross-validation approach.

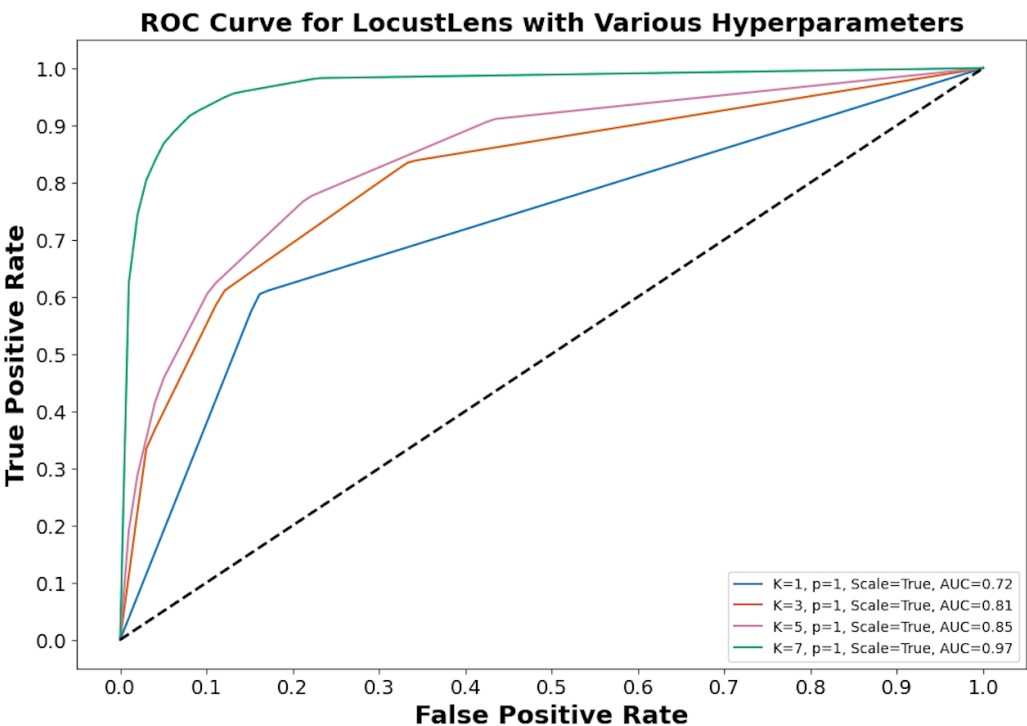

**Figure 8  LocustLens performance on different hyperparameters.**

LocustLens was tested under different configurations. For instance, with $k = 7$, weight as distance, and $p = 1$, the model delivered 96% accuracy, 0.97 recall, and a 0.97 F1-score. Using $k = 5$, weights as uniform, and $p = 2$, LocustLens produced 97% accuracy and 0.96 recall. However, when the value of k was reduced to 3 with uniform distance metrics and $p = 2$, accuracy dropped to 95%. Finally, after rigorous hyperparameter tuning, we concluded that $k = 7$, uniform weights, and the Manhattan distance provided optimal results, achieving 98% accuracy and recall. Figure 8 shows the impact of hyperparameters on the performance of LocustLens.

In this study, predicting locust attacks is critical, and the most important performance metrics are sensitivity (recall) and accuracy. Various models were tested and compared based on their recall for both class 0 (no locust attack) and class 1 (locust attack). The baseline-K-NN model achieved a recall of 98% for class 0 and 95% for class 1, with an overall average of 98%. The Bagging with SVC model followed with a class 0 recall of 97% and class 1 recall of 89%, leading to an overall average recall of 94%. The Boosting model had an average recall of 92%, while SVC produced 96% recall for class 0 and 80% for class 1, averaging 92%. LR performed less effectively, especially for class 1 (63% recall), resulting in an overall recall of 84%. The DT model achieved 94% recall overall.

In comparison, LocustLens outperformed all other models, achieving a recall of 98% for class 0 and 96% for class 1, with the highest overall average recall of 98%. Additionally, LocustLens demonstrated the highest accuracy and shortest execution time, making it the most efficient and robust model for predicting locust attacks across 42 countries. These

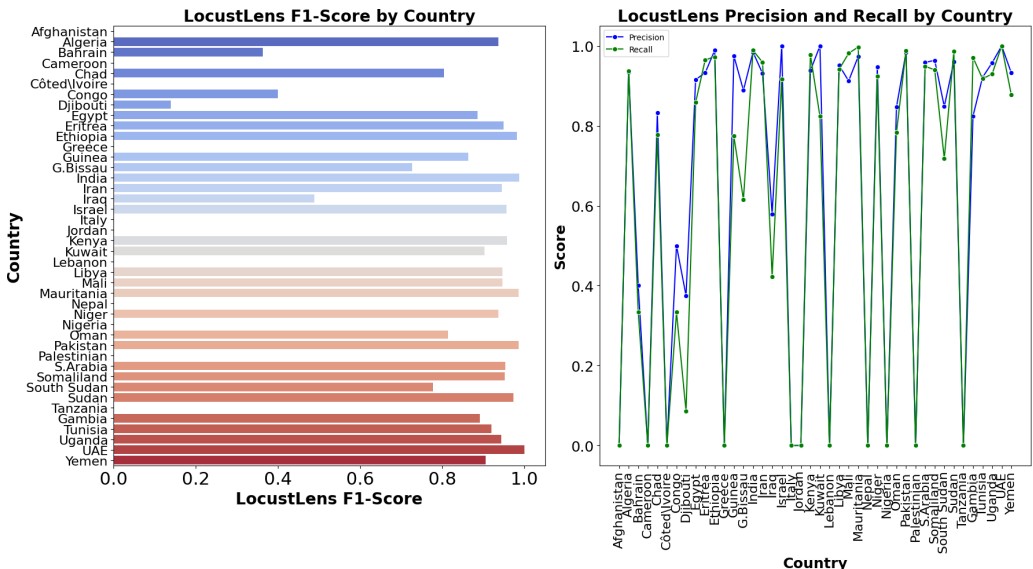

**Figure 9    Country-wise performance of LocustLens.**

results confirm that LocustLens is superior in terms of both sensitivity and computational efficiency when compared to traditional machine learning models. Since 42 countries are considered in this research the methodology of LocustLens constitutes a divide-and-conquer approach that tests the model on each country. The performance of LocustLens for each country is shown in Fig. 9. For top countries with sufficient data on locust attacks, the model gave 98% accurate predictions.

Since LocustLens does not require training, it recorded the lowest fitting and testing times at 0.009 and 0.001 s, respectively. In contrast, SVC had the highest training time at 1787.4 s and a testing time of 16.02 s. Baseline-K-NN fitting took 142.0 s, and its testing time was 1.598 s. DT and LR demonstrated slightly lower testing times at 0.0019 and 0.0014 s, respectively. A summary of all models' performance metrics—including accuracy, precision, recall, AUC, and F1-score—can be found in Table 7.

Environmental factors, particularly maximum temperature, soil moisture, and precipitation, were identified as significant contributors to locust infestations. The impact of these variables is reflected in class predictions, with maximum temperature emerging as a primary factor. Additionally, to evaluate the sustainability of our models, we measured their carbon emissions. Thus, we evaluated carbon emissions from ML models for a sustainable environment. We have used Google Colaboratory, where the assigned total CPU power is 42.5W, and available RAM is 12.678 GB. Our notebook launched in the US region. Regional carbon intensities for electricity grid in units of $CO_2$ equivalent per kilowatt-hour ($gCO_2eq/kWh$) are noted from *Electricity Maps (2023)* and *CodeCarbon (2023)* for the aforementioned region.

Equation (7) is used to measure carbon emissions and results are shown in Table 8.

The graphical performance of the LocustLens along with comparison models is displayed in Fig. 10. While evaluating the results, we carefully examined the frequency distribution

**Table 7  Performance comparison of LocustLens with other ML models.**

| Model | Accuracy | F1-score | AUC | Precision | Recall | Train time (secs) | Test time (secs) |
|---|---|---|---|---|---|---|---|
| LocustLens | 0.98 | 0.97 | 0.97 | 0.97 | 0.98 | 0.009 | 0.001 |
| Base-line K-NN | 0.96 | 0.96 | 0.96 | 0.97 | 0.98 | 142 | 1.59 |
| SVC | 0.91 | 0.89 | 0.88 | 0.89 | 0.92 | 1787.4 | 16.2 |
| DT | 0.97 | 0.97 | 0.96 | 0.96 | 0.94 | 3.07 | 0.0019 |
| LR | 0.83 | 0.79 | 0.77 | 0.75 | 0.84 | 5.154 | 0.0014 |
| AdaboostClassifier | 0.91 | 0.89 | 0.89 | 0.91 | 0.92 | 0.93 | 0.031 |
| Bagging with SVC | 0.94 | 0.93 | 0.92 | 0.94 | 0.94 | 196.50 | 75.28 |

**Table 8  Carbon footprints of ML models.**

| Model | Energy consumed (RAM) (kWh) | Energy consumed (CPU) (kWh) | Carbon emissions (kg CO2eq) |
|---|---|---|---|
| LocustLens | 0.0012 | 0.0111 | 4.870 |
| Base-line K-NN | 0.0012 | 0.0326 | 13.38 |
| SVC | 0.0023 | 0.0212 | 9.306 |
| DT | 0.00002 | 0.00018 | 0.079 |
| LR | 0.00003 | 0.0003 | 0.130 |
| Adaboost | 0.000007 | 0.000059 | 0.024 |
| Bagging Classifier | 0.001793 | 0.016036 | 0.650 |

of the 'yes' and 'no' classes concerning environmental factors, namely soil moisture, precipitation, and maximum temperature, as shown in Fig. 6. In addition, we have showcased the results obtained from various models, and it is evident that the LocustLens outperforms the others in accuracy, AUC, F1-score, precision, and recall.

# CONCLUSION

This study introduces LocustLens, a novel machine-learning model designed for global locust swarm prediction. The model integrates key environmental factors like soil moisture, temperature, and precipitation with locust presence data from 42 countries to create a robust and adaptable prediction tool. LocustLens is not only highly accurate, with excellent classification metrics such as precision, recall, and AUC, but it also features low fitting and testing times, making it efficient and easy to update with new data. In terms of adaptability locust attack data for new countries or regions can be incorporated into it without the need of extensive retraining and re-evaluation. Its integration of crucial environmental factors and ease of updating with new data make it a valuable tool for supporting agricultural decision-making, especially in regions vulnerable to food crises due to locust infestations. However, as we continue to apply machine learning models at a global scale, it is essential to address the environmental impact of these technologies.

While LocustLens demonstrates strong predictive capabilities, several limitations and challenges should be considered to provide a more balanced view. The GLAD42 dataset has limited samples for some countries, which could affect the model's generalizability,

**Performance Metrics of LocustLens and Comparison Models**

**Figure 10** Graphical performance evaluation of ML models.

necessitating improvements in data quality and sampling methods for broader applicability. Various combinations of encoder and decoder methods can be employed to enhance data sampling and address class imbalances. Moreover, the model cannot focus on separate regions due to the limited number of samples. It is recommended for future work to enhance data quality and samples so LocustLens can be applied for each country and region. Additionally, as the model's predictions rely on current environmental factors, the ongoing and unpredictable impacts of climate change could require frequent updates and recalibrations to maintain accuracy. Moreover, while LocustLens ranks third in terms of low carbon emissions, the cumulative environmental impact of deploying machine learning models at a global scale remains a concern. Future research should focus on enhancing data collection and developing databases that can forecast environmental changes to address these challenges effectively. Moreover, future research should also, focus on developing algorithms that not only enhance predictive capabilities but also promote environmental sustainability. Such efforts can help mitigate the adverse effects of climate change while fostering eco-friendly practices to conserve natural resources.

### Funding

This work was supported by FCT - Fundação para a Ciência e Tecnologia, I.P. by project reference UIDB/50008/2020, and DOI identifier https://doi.org/10.54499/UIDB/50008/2020.

The funders had no role in study design, data collection and analysis, decision to publish, or preparation of the manuscript.

## Grant Disclosures

The following grant information was disclosed by the authors:

FCT - Fundação para a Ciência e Tecnologia, I.P. by project reference UIDB/50008/2020, and DOI identifier https://doi.org/10.54499/UIDB/50008/2020.

## Competing Interests

Ivan Miguel Pires is Academic Editor for PeerJ Computer Science.

## Author Contributions

- Sidra Khan conceived and designed the experiments, performed the experiments, analyzed the data, performed the computation work, prepared figures and/or tables, authored or reviewed drafts of the article, and approved the final draft.
- Beenish Ayesha Akram conceived and designed the experiments, performed the experiments, analyzed the data, performed the computation work, prepared figures and/or tables, authored or reviewed drafts of the article, and approved the final draft.
- Amna Zafar conceived and designed the experiments, performed the experiments, analyzed the data, performed the computation work, prepared figures and/or tables, authored or reviewed drafts of the article, and approved the final draft.
- Muhammad Wasim conceived and designed the experiments, performed the experiments, analyzed the data, performed the computation work, prepared figures and/or tables, authored or reviewed drafts of the article, and approved the final draft.
- Khaldoon S. Khurshid conceived and designed the experiments, performed the experiments, analyzed the data, performed the computation work, prepared figures and/or tables, authored or reviewed drafts of the article, and approved the final draft.
- Ivan Miguel Pires conceived and designed the experiments, performed the experiments, analyzed the data, performed the computation work, prepared figures and/or tables, authored or reviewed drafts of the article, and approved the final draft.

## Data Availability

The FAO dataset is available at https://locust-hub-hqfao.hub.arcgis.com/.

The TerraClimate Dataset is available at https://climatedataguide.ucar.edu/climate-data/terraclimate-global-high-resolution-gridded-temperature-precipitation-and-other-water.

GLAD42 Dataset is available at Zenodo: Khan, S. (2024). GLAD42 - Global Locust Attack Dataset for 42 countries (Version 1) [Data set]. Zenodo. https://doi.org/10.5281/zenodo.13780945.

## Supplemental Information

Supplemental information for this article can be found online at http://dx.doi.org/10.7717/peerj-cs.2420#supplemental-information.

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
