# Peer review of "LocustLens: leveraging environmental data fusion and machine learning for desert locust swarm prediction"

_PeerJ Computer Science, doi:10.7717/peerj-cs.2420_

## Round 0.1 · original submission · Major Revisions

Thank you for submitting your manuscript to PeerJ Computer Science. We have now received and carefully considered the feedback from the reviewers.

After thorough evaluation, the reviewers have raised significant concerns regarding the methodology and the core proposal of your study. Specifically, they highlighted issues related to the robustness and clarity of your methodological approach, as well as potential limitations in the novelty and impact of your proposed solution.

Given the extent of these concerns and the critical nature of the feedback, we regret to inform you that we are unable to accept your manuscript for publication in its current form. We encourage you to address the reviewers’ comments comprehensively, which may require substantial revisions to your study design and approach.

We recognize the effort that went into your research and hope that you find the reviewers’ feedback constructive for further refinement of your work. We welcome you to resubmit your manuscript after addressing these concerns.

·

Basic reporting

The article titled 'LocustLens: Leveraging environmental data fusion and machine learning for desert locust swarm prediction' depicts an important problem of binary classification of whether a locust swarm attack occurs, given the environmental conditions. The text is interesting and clear; however, several issues should be addressed.

Experimental design

(1) The main strength is the new database, built from several sources. Please extend the description (section 4).
- Please title this section 'Dataset XXX description', not 'Brief description of dataset', where XXX is the dataset name.
- Please provide open access to the database to make revision possible
. Please add database statistics (tables), e.g., how many records are from each region/country, statistics on attributes, etc.

Moreover:

The addition of the negative example to the database is not an augmentation process; it should be called the generation.
Please report how many negative examples were added and provide statistics, e.g., how many negative examples were added for each country/region.

Text in the section '5 Methodology' also includes a database description, e.g., the number of records. Please do not repeat the same information.

Validity of the findings

(2) The new algorithm, LocusLength, should be examined more precisely.
- Please depict the differences between the proposed approach and others based on K-NN, e.g. Pusaden (2022) with an accuracy 99.3%;
- Please examine the influence of algorithm parameters on the results.

Additional comments

(3) Please reduce known information and not-relevant text; section 5.4 depicts known algorithm Support Vector Classifier, Decision Tree, Logistic regression, AdaBoost and Bagging. Please remove this part - it is not new and has no connection with the new method provided.

(4) Please reduce the description of evaluation metrics: confusion matrix, True Positive, ..., Accuracy, ..., AUC is basic knowledge, and there is no need to repeat definitions in the presented text. Section 6 should be reduced.

(5) Minor issues:

- please explain the KISS acronym, line 118
- space on page 6

·

Basic reporting

Good concept

Experimental design

Designed well

Validity of the findings

Good

·

Basic reporting

Should Sections 5.5, 5.6, 5.7 be under "Comparison Models" instead of their own sections, and should these be numbered 5.4.2, 5.4.3, 5.4.5 instead?

Experimental design

1) It'll be good to also show comparison with more advanced classifiers (like ResNet50, CNN+LSTM+FFN) mentioned in Literature Review, or add discussion about why the proposed approach is expected to perform better than those.

2) The authors can consider adding country wise performance for LocustLens? Are there any particular geographies where the proposed approach tends to perform better? And if so, what could be the reason for this?

3) Similar to #2, does the performance of proposed change with changes in variables like Tmax?

Validity of the findings

1) Fig 6 is a bit unclear, for example, in the image on the top left, what is the X axis represent? Can the axes be labeled more clearly to avoid confusion?

2) In Lines 474-475, instead of "while DT has 0.97%, baseline K-NN has 0.96%,
475 SVC approximates 0.91%, and LR has 0.83%", it should be "while DT has 97%, baseline K-NN has 96%, SVC approximates 91%, and LR has 83%". Same for line 476.

·

Basic reporting

1. Line # 9, Page 4 (also in abstract line # 24). Please describe the acronym before using it. What is FAO database?
2. Line # 25-26 - The term "negative instances in connection to all positive classes" is somewhat unclear. It would be beneficial to explain briefly what this means or rephrase it for better understanding.
3. The abstract mentions "reverse geocoding" but does not clearly explain these concepts or their relevance. A brief clarification of how reverse geocoding contributes to the methodology would help.
4. Although LocustLens is compared with several other models, the text does not provide details on why the other models were chosen or how their comparison contributes to understanding the performance of LocustLens. A brief mention of the criteria for model selection could be helpful.
5. The text does not discuss any limitations or potential challenges of the proposed methodology. Addressing limitations could provide a more balanced view and set realistic expectations for readers.
6. Literature references - For several papers, the dataset is mentioned but not fully detailed (e.g., Worldclim2, NOAA). A brief description of each dataset’s relevance or characteristics could provide better context.

Experimental design

1. The accuracy values provided for different models appear to be unusually high (e.g., 98% for LocustLens). This raises questions about the validity and source of these figures. Providing context on how these results were obtained and whether they were cross-validated or tested on a separate dataset would help.
2. The paper mentions that LocustLens outperforms other models, but the numbers provided for the accuracy, F1-score, and AUC are very close between models. This makes it hard to determine the actual significance of these differences. A clearer explanation of why LocustLens is considered superior despite these small differences would be beneficial.
3. The metrics for AdaBoostClassifier and Bagging with SVC are presented without context or explanation. It's not clear why these models are included, how they were chosen, or what their relevance is to the comparison.

Validity of the findings

1.Conclusion -
1. a - The text reiterates certain points, such as the effectiveness of LocustLens in terms of accuracy, precision, and efficiency, and its low carbon emissions. This repetition could be streamlined to avoid redundancy.
1.b - The future research directions are somewhat vague and could be more specific. For example, suggesting specific methods for how to integrate new data or more precise goals for improving predictive accuracy would be more actionable.
1.c - The claim of "global applicability" might be too ambitious without concrete evidence or case studies demonstrating effectiveness in various global regions outside the 42 countries analyzed.

---

## Round 0.2 · accepted · Accept

After carefully reviewing the revisions you have made in response to the reviewers' comments, I am pleased to inform you that your manuscript has been accepted for publication in PeerJ Computer Science.

Your diligent efforts to address the reviewers’ suggestions have significantly improved the quality and clarity of the manuscript. I confirm that the changes you implemented have successfully resolved the concerns raised, and the content now meets the high standards of the journal.

Thank you for your commitment to enhancing the paper. I look forward to seeing the final publication